# A Review on the Efficacy and Safety of Nab-Paclitaxel with Gemcitabine in Combination with Other Therapeutic Agents as New Treatment Strategies in Pancreatic Cancer

**DOI:** 10.3390/life12030327

**Published:** 2022-02-22

**Authors:** Christian Chapa-González, Karina López, Kimberly Michelle Lomelí, Jorge Alberto Roacho-Pérez, Jazmín Cristina Stevens

**Affiliations:** 1Grupo de Investigación de Nanomedicina-UACJ, Instituto de Ingeniería y Tecnología, Universidad Autónoma de Ciudad Juárez, Ciudad Juárez 32310, Mexico; al168494@alumnos.uacj.mx (K.L.); al181536@alumnos.uacj.mx (K.M.L.); 2Departamento de Bioquímica y Medicina Molecular, Facultad de Medicina, Universidad Autónoma de Nuevo León, Monterrey 64460, Mexico; alberto.roachoprz@uanl.edu.mx; 3Instituto de Ciencias Biomédicas, Universidad Autónoma de Ciudad Juárez, Ciudad Juárez 32310, Mexico

**Keywords:** pancreatic cancer, pancreas adenocarcinoma, drug combination, paclitaxel, gemcitabine, chemotherapy, drug response, overall survival, clinical trial

## Abstract

Pancreatic cancer has one of the highest mortality rates among cancers, and a combination of nab-paclitaxel with gemcitabine remains the cornerstone of first-line therapy. However, major advances are required to achieve improvements in patient outcomes. For this reason, several research groups have proposed supplementing treatment with other therapeutic agents. Ongoing studies are being conducted to find the optimal treatment in a first-line setting. In this work, we used a search strategy to compare studies on the efficacy and safety of nab-paclitaxel with gemcitabine in combination with other therapeutic agents based on the criteria of the Preferred Reporting Items for Systematic Reviews. We found seven studies in different clinical phases that met the inclusion criteria. The seven therapeutic agents were ibrutinib, necuparanib, tarextumab, apatorsen, cisplatin, enzalutamide, and momelotinib. Although these therapeutic agents have different mechanisms of action, and molecular biology studies are still needed, the present review was aimed to answer the following question: which formulations of the nab-paclitaxel/gemcitabine regimen in combination with other therapeutic agents are safest for patients with previously untreated metastatic pancreas ductal adenocarcinoma? The triple regimen is emerging as the first-line option for patients with pancreatic cancer, albeit with some limitations. Thus, further studies of this regimen are recommended.

## 1. Introduction

Pancreatic cancer continues to present challenges that have yet to be resolved by state-of-the-art medicine. The worldwide incidence of pancreatic cancer among men (5.7 per 100,000 people) is higher than that among women (4.1 per 100,000 people). This type of cancer is also the seventh leading cause of cancer death in both sexes and is more deadly in men (4.9 per 100,000 people) than in women (4.5 per 100,000 people) [1,2,3,4,5]. Adenocarcinoma of the exocrine pancreas represents 90% of pancreatic cancer cases, and its most widely accepted classifications are resectable, borderline resectable, and locally advanced pancreatic cancer [6,7]. The staging system used most often for pancreatic cancer is the TNM (tumor/node/metastasis) system from the American Joint Committee on Cancer (8th edition) [7,8,9,10,11]. Depending on the location of the tumor, most patients become symptomatic late in the disease. Consequently, patients with previously untreated advanced pancreatic ductal adenocarcinoma, who represent 50–55% of cases [7], have a very short life expectancy [12]. Therefore, efforts are currently being made to improve the diagnosis and treatment of this disease [13].

Recently, advances have been made to detect metastatic pancreatic ductal adenocarcinoma (PDAC) using molecular magnetic resonance imaging (MMRI) [14]. Additionally, advances have been made in liquid biopsy [15] and in finding specific biomolecular or subcellular targets [16,17,18,19], new therapeutic agents [20], and nanomedicine applications [21,22,23,24,25,26,27,28]. Moreover, our understanding of the molecular biology events of pancreatic cancer cells has increased [29,30,31]. At present, gemcitabine plus nab-paclitaxel is the preferred treatment for patients with pancreatic cancer [32,33,34,35]. Several research groups proposed complementing this treatment with other therapeutic agents seeking greater efficacy and safety. The advent of a therapy that decreases adverse events and improves overall survival outcomes in pancreatic cancer patient populations will be a milestone in medical research.

Currently, the indicated treatment is based on the stage and health status of the patient. Despite efforts to advance targeted therapy, immunotherapy [36], and nanomedicine [21,25], chemotherapy remains one of the most important therapeutic options, especially in PDAC. Over the last 30 years, treatment of PDAC has been improved from standard chemotherapies, consisting of fluoropyrimidines such as 5-FU and the antimetabolite drug gemcitabine, to new drug combinations. Adjuvant chemotherapy (after surgery) is the current method of care used for those with resectable pancreatic cancer, where gemcitabine as the single agent has a benefit in patient survival, particularly for those who have a limited functional state. However, regimens with multiple agents provide survival advantages, including gemcitabine plus capecitabine and FOLFIRINOX (fluorouracil, folinic acid, irinotecan, and oxaliplatin), which improves the disease-free survival over that with gemcitabine alone. However, this treatment is associated with higher toxicity [37,38].

The use of chemotherapy before surgery (neoadjuvant) to treat resectable cancer has uncertain benefits, but neoadjuvant chemotherapy has become the standard of care for some diseases, such as borderline resectable, locally advanced, and metastatic cancer. Locally advanced and metastatic diseases are treated with FOLFIRINOX or nab-paclitaxel with gemcitabine (NP/G). NP/G has shown good results in overall survival compared with gemcitabine monotherapy. In addition, progression-free survival and objective response rates were also improved [39,40,41]. Furthermore, when the efficacy and safety of NP/G and FOLFIRINOX were compared, the response rate was shown to be 6.3% in the FOLFIRINOX group and 40.9% in the NP/G group; drug toxicity in the NP/G group was also less than that in the FOLFIRINOX group [41].

Gemcitabine is a nucleoside analog of deoxycytidine and inhibits the progression of cells found in the G1/S phase. The intracellular uptake of gemcitabine is mediated mainly by nucleoside transporters (ENTs), while the unidirectional transport of nucleosides into cells is mediated by the family of concentrative nucleoside transporters (CNTs). For many years, gemcitabine monotherapy remained the gold standard of treatment for advanced PDAC. Then, abraxane and albumin-bounded paclitaxel nanoparticles (nab-paclitaxel) in combination with gemcitabine (NP/G) emerged as a new method of treatment for patients with metastatic pancreatic cancer [42]. Nab-paclitaxel was approved in 2013 for advanced-stage pancreatic cancer [43]. Paclitaxel is a widely used and successful natural antineoplastic drug that acts by stabilizing microtubules (polymers composed of repeated subunits of α- and β-tubulin heterodimers), increasing cell polymerization and stopping the cell cycle in the G2/M phase, which leads to cell death [40,44,45]. A combination of paclitaxel and other therapeutic agents was also shown to be effective, e.g., when used with palbociclib in triple-negative breast cancer (TNBC) [46]. Nab-paclitaxel is a formulation of paclitaxel with albumin that is synthetized via the homogenization of serum albumin at a concentration from 3 to 4%, with paclitaxel added to improve the drug’s biodistribution [44].

In the present study, we used a scoping review approach to compare the efficacy and safety of nab-paclitaxel with gemcitabine in combination with other therapeutic agents of interventions published in the literature based on the criteria of the Preferred Reporting Items for Systematic Reviews (PRISMA) extension for scoping reviews. This review was aimed to answer the following question: which of the formulations of the nab-paclitaxel/gemcitabine regimen in combination with other therapeutic agents are safest for patients with metastatic pancreas ductal adenocarcinoma? Findings from this review will be informative for researchers seeking to prioritize future advances in the treatment of pancreatic cancer.

## 2. Methods

### 2.1. Search Strategy

This review was guided by the protocol published in [47]. A systematic search was carried out on ScienceDirect, PubMed, and EBSCO Host during the period from 2015 to 2021. The search strategy used the following query: (nab-paclitaxel OR “nanoparticle albumin-bound paclitaxel”) AND (“Pancreatic Cancer” OR “Metastatic Pancreatic Cancer” OR “Pancreatic adenocarcinoma”) AND (Gemcitabine) AND (Chemotherapy) AND (“Clinical trial”). Once the search was completed, the articles were compiled, and two authors (C.C. and K.L.) independently assessed titles/abstracts for trial eligibility using a priori selection criteria. The final list of records was exported to an Excel file to delete the duplicates. The same authors separately evaluated the admissibility of the retrieved full-text trials. Consensus was reached by discussion.

### 2.2. Eligibility Criteria

We included records of clinical trials that involved patients over 18 years old with advanced or metastatic pancreatic cancer. Records were included if they were approved by an ethics committee. Records in which the NP/G regimen was combined with other therapeutic agents for the treatment of pancreatic cancer had to clearly indicate the dose used, adverse events, overall survival, and progression-free survival to be included in this review. In addition, the records had to specify some of the following criteria: complete response, partial response of the treatment, or objective response rate. Furthermore, the records were dismissed when they did not contain the main NP/G regimen, if another type of cancer was evaluated, or if the publication reported on a previous phase of the clinical study by the same authors or working group.

### 2.3. Study Selection

Initially, bibliographic information, titles, and abstracts were reviewed for eligibility. We removed articles that were not clinical trials or had publication dates before 2015. After full text screening, articles that did not address NP/G treatment in combination with other medications for the treatment of pancreatic cancer were discarded. Duplicates, if any, were then removed.

### 2.4. Data Extraction

From the studies included in this review, the following data were extracted: author/year/reference, number of patients, drugs and dose used, overall survival (OS), progression-free survival (PFS), partial response (PR), complete response (CR), objective response rate (ORR), and adverse events (AE) in any grade. For data recording, the studies were divided into two types: studies that made use of placebos (NP/G + placebo) and studies that did not employ placebos.

### 2.5. Quality Assessment of Included Studies

Reliability was evaluated using the Jadad tool [48,49,50] for randomized clinical trials. This tool employs a scale of 0–5 points; a score under 3 is equivalent to a low-quality record. The evaluation herein involved 5 questions: questions 1, 3, and 5 had values of one point each, and questions 2 and 4 were used to achieve higher or lower score levels, with values from 0 to −1. The nonrandomized clinical trials were evaluated using a tool for quasiexperimental studies developed by the Johanna Briggs Institute [51]. Finally, the PRISMA ScR checklist [52] was used to verify that the indicated items were met.

## 3. Results

### 3.1. Selection of Sources of Evidence

Out of the initial 1017 studies (after 194 duplicates were removed), 185 studies met the inclusion criteria and were included to form the basis of the analysis. Most of the studies were excluded (151) in the title and abstract screening stage because they were not relevant to the topic under review. Studies of treatments other than the NP/G combination were excluded. Figure 1 shows the PRISMA flowchart that describes the selection of sources of evidence.

### 3.2. Characteristics of Sources Evidence 

Four out of seven records identified were randomized clinical trials that employed placebos in the study [53,54,55,56], with 425 patients treated using placebos and 428 treated using the corresponding drug in combination with NP/G. One study was a phase III clinical trial [53], and the other three studies were phase II clinical trials [54,55,56]. In the same way, data from a pilot study and two phase I clinical trials [57,58,59] were collected. In these nonrandomized studies, the authors investigated the maximum tolerated dose of the additional drug in the NP/G regimen, and 74 patients participated. The treated disease was defined as metastatic PDAC in all studies, except in [59], where it was defined as metastatic or unresectable pancreatic cancer. From a total of 923 patients, 523 were men, and 400 were women. In [58], the mean age was 60.7 ± 10.2 years, while in the other studies, the age range was between 61 and 68 years (Table 1).

### 3.3. Critical Appraisal within Sources of Evidence 

The Jadad scale was applied to assess the quality of the studies (Table 2). One study [53] achieved a score of 4, two [54,55] achieved scores of 3, and one [56] achieved a score of only 2 out of a possible 5. Randomization and blinding issues were also identified. In the same way, the Joanna Briggs quasiexperimental evaluation protocol was applied (Table 3). Question 4 did not apply to any record, as the studies evaluated did not use control groups and were instead focused on finding the maximum tolerated dose of the additional drug in NP/G. 

### 3.4. Results of Individual Sources of Evidence 

The results obtained from each study included in this work are presented below. Table 4 and Table 5 summarize information on the treatment survival, response, and safety in the studies that involved placebos. The data are ordered from highest to lowest ORR according to the drug used in the research and not the placebo. Notably, in [53,55], the OS, PFS, and ORR were higher in NP/G plus placebo formulations. This result was also reported in [56] for the OS and PFS. However, the ORR was the same for NP/G plus drug.

The Common Terminology Criteria for Adverse Events (CTCAE) grade is based on the severity of an AE according to the Severity Grading Scale. The AE data extracted and shown in Table 5 and Table 6 correspond to all grades and patients reported in the articles. However, in this section, we compare some concise data from specific grades. In [53], it was observed that the AE with the highest percentages in drug plus placebo formulations were asthenia (16% vs. 12%), anemia (16% vs. 17%), neutropenia (24% vs. 35%), diarrhea (14% vs. 9%), and peripheral sensorial neuropathy (17% vs. 8%). 

In [54], there were more adverse events found in the group in which necuparanib was used than in the placebo group. In the third-grade AE, the use of NP/G plus necuparanib presented thrombocytopenia (27% vs. 5%), anemia (22% vs. 1%), fatigue (13% vs. 11%), elevated ALT (alanine aminotransferase) (12% vs. 0%), elevated AST (enzyme aspartate aminotransferase) (10% vs. 2%), and diarrhea (12% vs. 0%), while in the fourth-grade AE, two events were observed in a higher percentage of affected patients in the group with placebo: neutropenia and febrile neutropenia. For the fifth-grade AE, 12% and 9% of patients were found to be affected in the necuparanib and placebo groups, respectively. However, only 6% and 2% were registered. In [55], AE were present in a higher percentage of affected patients in the tarextumab group, except for neutropenia. However, in [56], adverse events were reported in a generalized way, with hypokalemia highlighted with a percentage of 17%.

In the same way, we compared studies that did not use a placebo. Notably, the authors in [57] presented very high ORR, OS, and PFS compared with those in [58,59]. In [57], patients were treated with different doses (25, 37.5, and 50 mg/m^2^) to determine the maximum tolerated dose. There was no record in this study of a CR, PR, or ORR for each dose; only the maximum tolerated dose was reported. 

In contrast, in [58], there was a maximum dose of 200 mg of momelotinib administered twice a day, with no CR or PR provided. However, the administration of 200 mg of momelotinib once a day yielded a higher ORR than the other doses administered in the study. OS and PFS were evaluated in 25 patients, and CR and PR were evaluated in 24 patients. Meanwhile, in [59], two doses of enzalutamide were administered in phase 1a. The CR and PR here did not indicate which doses they corresponded to. In phase 1b, the recommended dose of 160 mg was administered to patients with androgen receptor expression in the tumor. In [57,58,59], only one datum was reported for OS and PFS in all studies (Table 4).

## 4. Discussion

### 4.1. Summary of Evidence

The objective of this review was to provide an update on the different formulations of NP/G in combination with other therapeutic agents to treat advanced PDAC and to determine which of these formulations is the safest for treatment of this disease. Ultimately, only seven studies fulfilled the eligibility criteria. Four out of the seven studies addressed the use of a placebo. Three of these studies were randomized phase II trials, and one was a phase III trial, which investigated, respectively, if the treatment was efficient and if the formulation was better than a conventional treatment. Table 7 shows the diversity of therapeutic agents under investigation in clinical trials. Considering the diversity in the chemical structures of the therapeutic agents enlisted, molecular biology studies will likely be needed to relate cellular or molecular events with the responses of patients to the triple regimen. 

The molecular effects of paclitaxel, gemcitabine, and cisplatin are well characterized in cancer cells [60,61,62,63,64,65,66,67,68]. However, a successful triple regimen whose cellular events are known with certainty will bring about a new paradigm for the treatment of pancreatic cancer. Clinical trials of combination therapies that are effective and safe should be complemented by molecular studies to understand the pathways for their biological activities.

The MPACT, a randomized phase III study, reported that NP/G had an OS of 8.5 months, a PFS of 5.5 months, a CR of less than 1%, and an RP of 23% in 431 patients, resulting in greater efficacy than gemcitabine monotherapy. The AE of third grade or higher were as follows: neutropenia (38%), fatigue (17%), and neuropathy (17%). Febrile neutropenia was also present in 3% of the patients [83], whereas in [53,55,56], placebo in combination with the NP/G regimen exceeded the formulation of the main regimen with the additional drug in ORR, OS, and PFS, which means that adding ibrunitib (Bruton’s tyrosine kinase inhibitor), tarextumab (IgG2 antibody against Notch2 and Notch3 receptors), or apatorsen (antisense oligonucleotide targeting heat shock protein 27 messenger RNA) was not more effective than the standard therapy of NP/G. While the formulation with necuparanib (heparin mimetic) was slightly superior to the placebo formulation, there was no significant improvement in OS and PFS, resulting in the same ORR as the NP/G standard therapy.

Regarding safety, the highest incidence of affected patients was observed with the NP/G formulation plus the investigated drug in [55]. Fourth- and fifth-grade AE had higher incidences in [54]. However, the AE in [53] and the third-grade AE in [54] did not differ considerably between the formulation with placebo and the formulation with the drug. In [56], the AE were reported in a general way; notably, this study had a poor-quality methodology and, therefore, a high risk of bias, conferring small reliability to the results.

The three remaining studies dealt with a pilot trial and two phase II trials. The objective was to detect possible improvements in the NP/G regimen plus drug and determine the maximum tolerated dose in which the tumor had response signals to the administered formulation; ultimately, no adverse events were observed. In [57], cisplatin (a cytotoxic chemotherapy agent) had ORR, PFS, and OS higher than those described in [83]. Hematologic toxicity was also frequently observed but did not need clinical intervention, as an acceptable safety profile was obtained. However, since the study was not randomized, there remains a risk of bias. 

At the same time, in [58], the OS and PFS did not have notable differences from data from the MPACT trial. With a dose of 200 mg once a day, there was a high ORR, but this ORR did not exceed the ORR in [57]. Well-tolerated AE were observed, but with a high presence of neuropathy due to nab-paclitaxel and momelotinib (an agent with inhibitory activity of Janus kinase 1 and 2). Furthermore, in [59], by using enzalutamide (an androgen receptor antagonist), the OS and the PFS were found to be higher than those reported in [58], with a superior ORR in phase 1a of the study. The toxicities that were most frequently reported in this study were hematological and gastrointestinal AE, which means that the safety was generally acceptable. Notably, 75% of the patients did not present an increase in tumor size with the formulation, and these patients also manifested androgen receptors in cancer cells. 

According to [57,59], a decrease of 90% in the level of carbohydrate antigen 19-9 (CA 19-9) was observed. This antigen is generally used as a biomarker, although it is usually unspecific. Therefore, confirmation of this clinical finding should be performed using the Response Evaluation Criteria in Solid Tumors (RECIST), which provides a simple and pragmatic methodology to evaluate the activity and efficacy of new cancer therapeutics in solid tumors using validated and consistent criteria to assess changes in tumor burden [84,85].

Finally, it is important to mention that drugs are not excluded solely because they cause adverse events. However, the benefits that the drugs provide to the body must outweigh the risks they can cause. Although the authors noted acceptable safety levels in each study, the efficacy that the studies presented in terms of survival and tumor response were not superior to those that were obtained in standard NP/G therapy, nor did they exceed the NP/G plus placebo formulation.

### 4.2. Limitations

Heterogeneity is an important challenge when reviewing clinical literature. There was substantial heterogeneity among the various studies in terms of NP/G formulations in combination with other therapeutic agents for cancer treatment. For example, the studies varied in their clinical phases, numbers of patients, doses administered, reports of adverse events, etc. Our results indicate the need for studies that compare the effects when administering different additional drugs to an NP/G regimen that includes similar doses and tumor stages in order to compare responses between treatments. In addition, our results show that it is important to reduce the risk of methodological bias, to choose proper inclusion and exclusion criteria for these trials, and to agree on the reportable results based on the quality of each individual study, which could provide results for subsequent analysis of data in reviews such as this one.

Our results should be interpreted while considering the limitations of the included studies due to very high risk of bias. More studies with controls are necessary to elucidate the benefits of including therapeutic agents to the NP/G regimen. These agents should improve therapeutic effects; reduce adverse events; and, above all, lead to reductions in or elimination of instances of pancreatic cancer in any of its stages.

## 5. Conclusions

The effective and safe treatment of pancreatic cancer represents a major challenge for medical research. One of the strategies that research groups test is the combination of therapeutic agents and their effectiveness. This review explored the 6-year progression of nab-paclitaxel with gemcitabine in combination with other therapeutic agents as new therapeutic strategies in pancreatic cancer. Therapeutic agents currently being studied include ibrutinib, necuparanib, tarextumab, apatorsen, cisplatin, enzalutamide, and momelotinib. Only NP/G+necuparanib achieved a greater variation in overall survival than the NP/G regimen, while NP/G+cisplatin regimen is emerging as a candidate for an effective therapeutic strategy, although the phase 1b/2 study still has limitations. More studies should be conducted to corroborate the benefits of adding other drugs to the NP/G formulation.

## Figures and Tables

**Figure 1 life-12-00327-f001:**
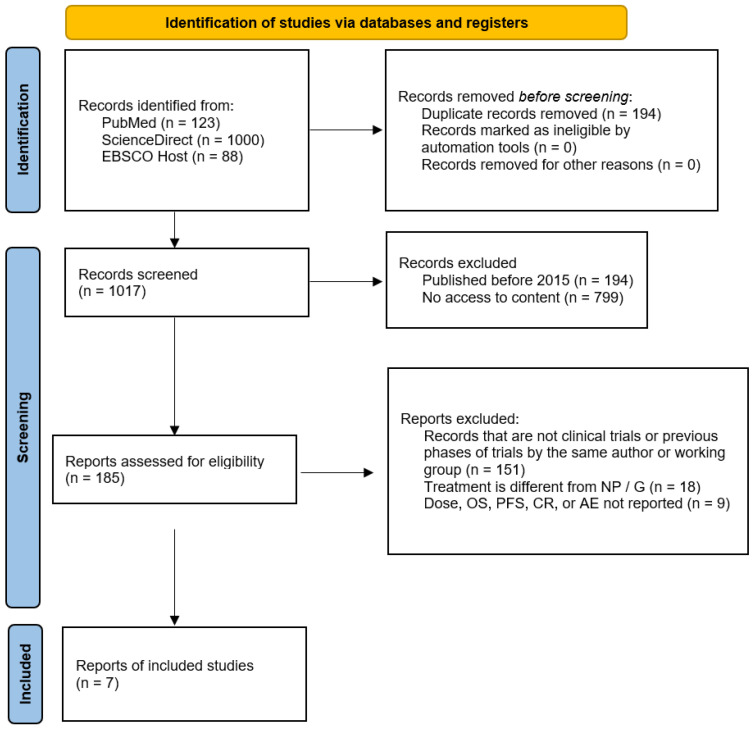
PRIMSA flowchart. The search strategy is reported here according to PRISMA guidelines. Abbreviations: AE, adverse events; CR, complete response; NP/G, nab-paclitaxel plus gemcitabine; OS, overall survival; PFS, progression-free survival.

**Table 1 life-12-00327-t001:** Patient characteristics.

Study	Clinical Phase	Sex		Age
		Male	Female	Median
M. Tempero et al. (2021) [53]	III	189	235	64
E. M. O’Reilly et al. (2020) [54]	II	58	62	64
Z. I. Hu et al. (2019) [55]	II	73	104	66
A. H. Ko et al. (2017) [56]	II	57	75	66
G. S. Jameson et al. (2020) [57]	1b/2	11	14	65
R. K. Mahipal et al. (2020) [58]	I	4	16	68
K. Ng et al. (2019) [59]	I	8	17	61

**Table 2 life-12-00327-t002:** The Joanna Briggs Institute Critical Appraisal Checklist.

	M. Tempero et al. (2021) [53]	E. M. O’Reilly et al. (2020) [54]	Z. I. Hu et al. (2019) [55]	A. H. Ko et al. (2017) [56]
Randomized	Yes	Yes	Yes	Yes
Appropriately randomized	No	No	No	No
Described withdrawals	Yes	Yes	Yes	Yes
Double-blinded	Yes	No	No	No
Described blinding	Yes	Yes	Yes	No
Jadad score	4	3	3	2

**Table 3 life-12-00327-t003:** The Joanna Briggs Institute critical appraisal checklist for quasiexperimental studies.

Study	Question								
	Q1	Q2	Q3	Q4	Q5	Q6	Q7	Q8	Q9
G. S. Jameson et al. (2020) [57]	Yes	Yes	Yes	Not applicable	Yes	Unclear	Yes	Unclear	Yes
R. K. Mahipal et al. (2020) [58]	Yes	Yes	Yes	Not applicable	Yes	No	Yes	Unclear	Yes
K. Ng et al. (2019) [59]	Yes	Yes	Yes	Not applicable	Yes	Yes	Yes	Unclear	Yes

Question 1. Is it clear in the study what is the “cause” and what is the “effect”? Question 2. Were the participants similar in any included comparisons? Question 3. Were the participants included in any comparisons receiving similar treatments/care, other than the exposure or intervention of interest? Question 4. Was there a control group? Question 5. Were there multiple measurements of the outcome, both pre- and post-intervention/exposure? Question 6. Was a follow up completed? If not, were differences between groups in terms of their follow ups adequately described and analyzed? Question 7. Were the outcomes of participants included in all comparisons measured in the same way? Question 8. Were outcomes measured in a reliable way? Question 9. Was appropriate statistical analysis used?

**Table 4 life-12-00327-t004:** Comparison of response to treatment.

Study	Patients n	Regimen *	OS, Months	PFS, Months	PR, n (%)	CR, n (%)	ORR
M. Tempero et al. (2021) [53]	213	NP/G + placebo	10.8	6	90 (42%)	3 (1%)	43
	211	NP/G + ibrutinib (560 mg)	9.7	5.3	62 (29%)	0	29
E. M. O´Reilly et al. (2020) [54]	58	NP/G + placebo	9.99	6.93	8 (14%)	2 (3%)	17
	62	NP/G + necuparanib (5 mg/kg)	10.71	5.52	14 (23%)	0	23
Z. I. Hu et al. (2019) [55]	88	NP/G + placebo	7.9	5.5	28 (32%)	0	32
	89	NP/G + tarextumab (15 mg/kg)	6.4	3.7	18 (20%)	0	20
A. H. Ko et al. (2017) [56]	66	NP/G + placebo	6.9	3.8	12 (18%)	0	18
	66	NP/G + apatorsen (600 mg)	5.3	2.7	12 (18%)	0	18
G. S. Jameson et al. (2020) [57]	25	NP/G + cisplatin (25 mg/m^2^)	16.4	10.1	15 (62.5%)	2 (8.33%)	71
A. Mahipal et al. (2020) [58]	12	NP/G + enzalutamide (80 and 160 mg)	9.73	7.53	4 (33%)	0	33
	12	NP/G + enzalutamide (160 mg)			1 (8.33%)	0	8.33
K. Ng et al. (2019) [59]	7	NP/G + momelotinib (100 mg once daily)	8.7	5.7	2 (28.6%)	0	29
	4	NP/G + momelotinib (150 mg once daily)			1 (25%)	0	25
	7	NP/G + momelotinib (200 mg once daily)			3 (42.9%)	0	43
	3	NP/G + momelotinib (150 mg twice daily)			1 (33.3%)	0	33.33

* Treatment consisted of the administration of the drug or placebo in combination with intravenous nab-paclitaxel (125 mg/m^2^) and gemcitabine (1000 mg/m^2^). OS = overall survival; PFS = progression-free survival; PR = partial response; CR = complete response; ORR = overall response rate.

**Table 5 life-12-00327-t005:** Summary of all grades of adverse events (AE) in treatments that included placebo.

Disorders	M. Tempero et al. (2021) [53]		E. M. O´Reilly et al. (2020) [54]		Z. I. Hu et al. (2019) [55]		A. H Ko et al. (2017) [56]
	Placebo, n = 212	Ibrutinib, n = 208	Placebo, n = 57	Necuparanib, n = 60	Placebo, n = 85	Tarextumab, n = 87	Apatorsen, n = 12
Abdominal pain	34%	32%	26%	25%	—	—	—
Alopecia	41%	43%	—	—	—	—	—
ALT increase	—	—	12%	35%	—	—	—
Anemia	45%	44%	—	—	26%	29%	17%
AST increase	—	—	16%	27%	—	—	—
Constipation	45%	49%	—	—	—	—	8%
Decreased appetite	37%	33%	—	—	13%	17%	17%
Dehydration	36%	41%	—	—	12%	9%	8%
Diarrhea	52%	71%	21%	50%	40%	72%	58%
Dysgeusia	20%	13%	—	—	9%	13%	8%
Dysphagia	—	—	—	—	—	—	8%
Dyspnea	31%	38%	—	—	—	—	—
Epistaxis	52%	56%	—	—	1%	10%	—
Fall	—	—	—	—	—	—	8%
Fatigue	40%	35%	54%	60%	59%	52%	42%
Fever	36%	29%	—	—	12%	9%	—
Hyperbilirubinemia	—	—	5%	3%	—	—	—
Hyperglycemia	—	—	—	—	—	—	8%
Hypersensitivity	—	—	—	—	—	—	8%
Hypokalemia	—	—	12%	15%	—	—	17%
Hypomagnesemia	—	—	—	—	—	—	8%
Hyponatremia	—	—	12%	22%	—	—	—
Hypophosphatemia	—	—	2%	10%	—	—	—
Insomnia	—	—	—	—	—	—	8%
Mucosal inflammation	—	—	—	—	—	—	8%
Myalgia	—	—	—	—	—	—	8%
Nausea	30%	33%	33%	53%	31%	41%	67%
Neuropathy peripheral	—	—	25%	18%	—	—	—
Neutropenia	6%	7%	—	—	18%	9%	—
Peripheral edema	41%	44%	21%	27%	—	—	8%
Peripheral embolism	—	—	—	—	—	—	8%
Pericardial effusion	—	—	—	3%	—	—	—
Peripheral sensory neuropathy	—	—	9%	8%	—	—	8%
Pleural effusion	—	—	2%	7%	—	—	—
Pneumonia	—	—	4%	7%	—	—	—
Pruritus	—	—	—	—	—	—	8%
Rash	—	—	—	—	—	—	24%
Sinus tachycardia	—	—	—	—	—	—	8%
Stomatitis	—	—	—	—	—	—	25%
Temperature intolerance	—	—	—	—	—	—	8%
Thrombocytopenia	26%	37%	—	—	25%	49%	17%
Vomiting	42%	42%	—	—	16%	22%	42%
Weight loss	—	—	—	—	—	—	8%

**Table 6 life-12-00327-t006:** Summary of all grades of adverse events (AE) in treatments that included no placebo.

Adverse Event	G. S. Jameson et al. (2020) [57] *	K. Ng et al. (2019) [59]	A. Mahipal et al. (2020) [58]
Abdominal pain	—	44%	41.67%
Acute cryptosporidiosis	yes	—	—
Alkaline phosphatase increase	—	—	66.67%
Alopecia	—	40%	20.84
ALT increase	—	—	58.34%
Anemia	yes	68%	91.67%
Anorectal infection	yes	—	—
Arthralgia	—	—	37.5%
AST increase	—	—	50.01%
Bilirubin increase	—	—	16.67%
Cachexia	—	4.00%	—
Constipation	—	52%	—
Cough	—	—	12.5%
Death	yes	—	—
Decreased appetite	—	40%	—
Decreased neutrophil count	yes	8.00%	58.34%
Decreased weight	—	4.00%	—
Deep vein thrombosis	—	4.00%	—
Dehydration	yes	4.00%	12.5%
Diarrhea	yes	64%	62.51%
Dizziness	—	—	12.5%
Dysgeusia	—	40%	—
Dyspnea	—	—	37.5%
Edema limbs	—	—	25%
Embolic stroke	—	4.00%	—
Epistaxis	yes	—	16.67%
Fall	—	—	12.5%
Fatigue	yes	80%	62.5%
Febrile neutropenia	yes	4.00%	—
Fever	yes	4.00%	41.67%
Generalized edema	—	—	—
Generalized muscle weakness	—	—	20.84%
Headache	—	—	20.84%
Hyperkalemia	—	—	29.17%
Hypertension	—	36%	16.66%
Hypoalbuminemia	—	—	45.84%
Hypokalemia	yes	—	20.83%
Hyponatremia	—	—	41.67%
Increased blood uric acid	—	4.00%	—
Lung infection	—	—	12.5%
Lymphocyte count decreased	yes	—	37.5%
Lymphocyte count increased	yes	—	—
Maculopapular rash	—	—	25%
Malaise	—	4.00%	—
Mucositis	—	—	25%
Myalgia	—	—	16.67
Nausea	yes	76%	70.84%
Nephrolithiasis	—	4.00%	—
Neutropenia	—	16%	—
Peripheral edema	—	48%	—
Peripheral motor neuropathy	yes	—	—
Peripheral neuropathy	—	36%	—
Peripheral sensory motor neuropathy	—	8%	54.17%
Peripheral sensory neuropathy	—	36%	—
Platelet count decreased	yes	—	70.84%
Pneumonia	—	24%	—
Polyneuropathy	—	4.00%	—
Pyrexia	—	56%	—
Respiratory distress	—	4.00%	—
Stroke	yes	—	—
Thrombocytopenia	—	8%	—
Thromboembolic event	—	—	20.83%
Tremor	—	4.00%	—
Urinary tract infection (UTI)	—	—	12.5%
Vomiting	yes	52%	50%
White blood cell decreased	yes	—	66.67%
Wound infection	—	—	8.33%

* The percentage among all patients was not indicated, since the authors reported the adverse effects as follows: “A patient who experienced multiple events within a system organ class (SOC) or preferred term was counted once for that class and once for the preferred term at the maximum observed grade”. Acronyms: ALT, alanine aminotransferase; AST, aspartate aminotransferase; UTI, urinary tract infection.

**Table 7 life-12-00327-t007:** Summary of treatment regimens for patients with previously untreated advanced pancreatic ductal adenocarcinoma (PDAC).

Ref	Therapeutic Agent	Structure	Description
[53]	Ibrutinib	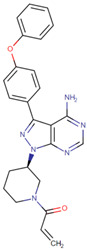	Ibrutinib is a Bruton’s tyrosine kinase inhibitor that forms a covalent bond with a cysteine residue (Cys 481). Ibrutinib is used to treat chronic lymphocytic leukemia, mantle cell lymphoma, and Waldenstrom‘s macroglobulinemia, leading to inhibition of BTK activity [69,70]. ClinicalTrials.gov identifier: NCT024366. Phase III RESOLVE study. Ibrutinib plus nab-paclitaxel/gemcitabine did not improve OS or PFS for patients with metastatic PDAC.
[54]	Necuparanib	—	Necuparanib (a heparin mimetic) acts as a multitargeting therapeutic, altering multiple signaling pathways simultaneously by binding and sequestering different proteins [71,72]. ClinicalTrials.gov identifier: NCT01621243. A randomized phase II trial. Necuparanib plus nab-paclitaxel/gemcitabine did not improve OS.
[55]	Tarextumab	—	Monoclonal antibodies (mAb, anti-Notch2/3, OMP-59R5) are fully human monoclonal antibodies that target the Notch2 and Notch3 receptors. They have been used in trials studying the treatment of solid tumors, stage IV pancreatic cancer, and stage IV small cell lung cancer [73,74]. ClinicalTrials.gov identifier: NCT01647828. A randomized phase II trial. Tarextumab plus nab-paclitaxel/gemcitabine did not improve OS, PFS, or ORR in first-line metastatic PDAC
[56]	Apatorsen	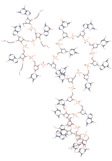	Apatorsen is a second-generation antisense drug in preclinical experiments that inhibits the production of heat shock protein 27 (Hsp27), a cell survival protein found at elevated levels in many human cancers, including prostate, lung, breast, ovarian, bladder, renal, pancreatic, multiple myeloma, and liver cancer [75,76]. ClinicalTrials.gov identifier: NCT01844817. A randomized, double-blinded, phase II trial. The RAINIER trial. Addition of apatorsen to nab-paclitaxel/gemcitabine regimen did not improve survival or other clinically relevant endpoints in patients with metastatic pancreatic cancer.
[57]	Cisplatin	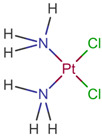	Cisplatin is a platinum-based chemotherapy agent used to treat various sarcomas, carcinomas, lymphomas, and germ cell tumors. Cisplatin exerts its anticancer activities by generating DNA lesions through interactions with purine bases, leading to the activation of various signal transduction pathways leading to apoptosis [68,77,78]. ClinicalTrials.gov identifier: NCT01893801. A nonrandomized phase 1b/2 pilot clinical trial. The addition of cisplatin to nab-paclitaxel/gemcitabine resulted in a high response rate and evolving OS.
[58]	Enzalutamide	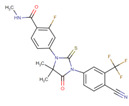	Enzalutamide is a rationally designed, targeted androgen-receptor inhibitor used to treat castration-resistant prostate cancer. Enzalutamide acts both by inhibiting the translocation of the androgen receptor into the nucleus and by reducing the transcriptional activity of this receptor [79,80]. ClinicalTrials.gov identifier: NCT02138383. A phase I trial. Enzalutamide plus nab-paclitaxel/gemcitabine was safely administered with no unexpected toxicities and resulted in consistent reductions in CA 19–9 (biological marker) levels.
[59]	Momelotinib	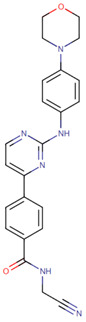	Momelotinib is a benzamide that acts as an ATP-competitive JAK1/JAK2 inhibitor. Momelotinib has been used in trials studying the treatment of polycythemia vera, primary myelofibrosis, post-polycythemia vera, essential thrombocythemia, and primary myelofibrosis (PMF), among others [81,82]. ClinicalTrials.gov identifier: NCT02101021. Phase 1 dose-escalation study. Momelotinib plus nab-paclitaxel/gemcitabine was safe and well tolerated, with no OS or PFS benefits.

## Data Availability

The datasets generated during and/or analyzed during the current study are available from doi:10.5281/zenodo.5855008 (accessed on 18 January 2022). Review Research Protocol. Efficacy and safety of nab-paclitaxel with gemcitabine in combination with other therapeutic agents in the treatment of pancreatic cancer. Available from: https://doi.org/10.5281/zenodo.4741412 (accessed on 18 January 2022).

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
