# Peer review of "A Review on the Efficacy and Safety of Nab-Paclitaxel with Gemcitabine in Combination with Other Therapeutic Agents as New Treatment Strategies in Pancreatic Cancer"

_life, 2022, doi:10.3390/life12030327_

Round 1

Reviewer 1 Report

“A review of the efficacy and safety of nab-paclitaxel with Gemcitabine in combination with other drugs as therapeutic strategies in pancreatic cancer” is a systematic review including publications from 2015 to 2021.

The authors aim to investigate which formulations of Nab-Paclitaxel/Gemcitabine regime in combination with other drugs have been safer for patients with pancreatic cancer.

I have a few comments that I have listed as minor (typos, style) or major (potential bias, major concern) in the order wherein they appear in the manuscript:

  1. GENERAL – The present paper would benefit from a native English speaker’s review for grammar and syntax repetitions (major)
  2. INTRODUCTION – The phrase from line 41 to line 45 is grammatically incorrect and I cannot understand what you mean exactly. Please rephrase (minor)
  3. INTRODUCTION – Line 55 – Did you mean “borderline”? (minor)
  4. INTRODUCTION – I believe Your introduction is too broad and I get lost reading your sentences. There are many syntax errors and there is no focus on the aim of your study (major)
  5. RESULTS – From your results, I do not understand which are the “other drugs” You are referring to throughout the manuscript (major)

Overall, the major limitation of the present study is the high heterogeneity of included studies. This descriptive systematic review does not seem to add to the currently available literature.

Author Response

Dear Reviewer, thank you for reading our manuscript and for sending us your comments and suggestions. Our responses are given in a point-by-point manner below. We have highlighted the changes to our manuscript within the document by using colored text in MS Word.

1. GENERAL – The present paper would benefit from a native English speaker’s review for grammar and syntax repetitions (major)

Response: Thank you for taking the time to carefully read the manuscript. The revised manuscript was sent to a professional proofreading service for English language. You will find the corrected version with the changes highlighted. In addition, we know that if the manuscript is accepted, it will go through the language edition again, this time by the editorial office, so the grammar and phrasing of the final version of the article will be improved.

2. INTRODUCTION – The phrase from line 41 to line 45 is grammatically incorrect and I cannot understand what you mean exactly. Please rephrase (minor)

Response: We agree with your observation. The phrase was originally Furthermore, in the development of safer and more effective pancreatic cancer treatments research is being carried out on and the combination of new therapeutic agents with gemcitabine and nab-paclitaxel, which remains the main first-line treatment options [25–27], there are even reports of exceptional response when it is used as a second-line treatment [28].

Nowadays, gemcitabine plus nab-paclitaxel is the treatment for patients with pancreatic cancer [25–28]. Several research groups propose to complement the treatment with other therapeutic agents seeking greater efficacy and safety  

3. INTRODUCTION – Line 55 – Did you mean “borderlined”? (minor)

Response: Yes, we meant "borderline resectable", however the paragraph was deleted, see next comment.

4. INTRODUCTION – I believe Your introduction is too broad and I get lost reading your sentences. There are many syntax errors and there is no focus on the aim of your study (major)

Response: Thanks for your observation. We have attended to it and, consequently, we have eliminated the second paragraph of the introduction because we reconsider that it is not focused on the main objective of the study. 

We have also removed the following text: Some of the symptoms of advanced pancreatic cancer are jaundice, abdominal pain, loss of appetite, pain in the back, epigastric pain, nausea, pancreatitis, involuntary weight loss, among others [29–31], for the same reason. The order of the references has been updated.

Syntax errors were corrected, you will find the changes made in the new version of the manuscript.

5. RESULTS – From your results, I do not understand which are the “other drugs” You are referring to throughout the manuscript (major)

Response: Thanks to your comment, we have reviewed the relevance of the term "other drugs". Because a therapeutic agent could be a small molecular weight drug or a macromolecule such as protein or DNA, we have changed the term "other drugs" to "other therapeutic agents" throughout the manuscript, including the title.

Overall, the major limitation of the present study is the high heterogeneity of included studies. This descriptive systematic review does not seem to add to the currently available literature.

Response: We respect your opinion, however we differ. This review does contribute to the literature. To the best of our knowledge, there is no published scientific article where the information on the treatments mentioned here is compiled and compared. 

Once again, we thank you for taking the time to carefully read our manuscript and for making all these observations. We hope that our responses are satisfactory and that you find the revised version of the manuscript suitable for publication.

Sincerely, 

Reviewer 2 Report

The authors have found 7 studies in different clinical phases that met the inclusion criteria: The 7 therapeutic agents listed are ibrutinib, necuparanib, tarextumab, apatorsen, cisplatin, enzalutamide, and momelotinib. Although therapeutic agents have different mechanisms of action and molecular biology studies are needed, this review aims to answer the following question: Which of the formulations of the Nab-Paclitaxel/Gemcitabine regimen in combination with other drugs has been safer for patients with pancreatic cancer? The triple regimen is emerging as the first-line option in patients with pancreatic cancer, although with some limitations, so further studies of the regimen are recommended. 

Pancreatic cancer is one of the leading causes of cancer death worldwide. Its high mortality rate has remained unchanged for years. Radiotherapy and surgery are considered standard treatments in early and locally advanced stages. Chemotherapy is the only option for metastatic patients. More studies are needed to provide an overview of prognostic and predictive markers used in clinical practice and to explore the most promising fields of research in terms of treatment selection and tailored therapy in pancreatic cancer. 

Any effort aimed at improving our knowledge of this very aggressive type of cancer must be valued and considered as an open window for its future cure.

However, the English should be reviewed to improve its content.

Author Response

Dear Reviewer, thank you for reading our manuscript and for sending us your comments and suggestions. 

Comments and Suggestions

  • Any effort aimed at improving our knowledge of this very aggressive type of cancer must be valued and considered as an open window for its future cure.

However, the English should be reviewed to improve its content.

Response: The English was proofread by a native speaker and any grammatical or syntax errors were corrected, in the updated version of the manuscript you will find the changes highlighted

Reviewer 3 Report

Manuscript ID: life-1582702.

Manuscript type: Systematic review.

A Review on the Efficacy and Safety of nab-paclitaxel with Gemcitabine 
in Combination with other Drugs as Therapeutic Strategies in Pancreatic Cancer

Christian Chapa, Karine López, Kimberly Michelle Lomelí, Jorge Alberto Roacho, Jazmín Cristina Stevens

Chapa C. et al. summarized in their systematic review recent data regarding the clinical effects of a combined treatment of gemcitabine plus nab-paclitaxel plus additional compounds in human pancreatic cancer. Their manuscript shows relevant information, particularly secondary effects, for physicians aiming to follow novel therapies for pancreatic cancer subjects. Although the manuscript deserves to be accepted, there are certain elements that have to be worked on before the acceptance.

  1. a) Major comments

  1. The authors must request a professional proofreading service for English language, due to the multiple typos and errors found thorough the text. Besides, the way some paragraphs are written is very suggestive of a literal translation from the Spanish language, without taking into account the grammatical rules of English. Thus, the authors are required to submit a proof of such service together to their next manuscript version. Some typos include, but are not limited to:

  1. a) However (line 71).
  2. b) Neadjuvant (line 72).
  3. c) The locally advanced or metastatic (line 74).
  4. d) , it (line 88).
  5. e) Tolerate (line 166).
  6. f) 74 patients participated (lines 166-167).
  7. g) Can, without the word ‘it’ before (line 205).
  8. h) Comparisons similar (Table 3).
  9. i) Edema peripheral (Table 5).
  10. j) Neuropathy peripheral (Table 5).
  11. k) Weight decreases, instead of loss (Table 5).
  12. l) Alkaline phosphatase increase (Table 6).
  13. m) An ‘of’ missed between ‘grades’ and ‘adverse’ (title of Table 6).
  14. n) All the ‘it’ that are missing at the beginning of the background paragraphs in Table 7.

Besides, some of the paragraphs are excessively long and required to be split in shorter paragraphs to avoid confusions in their reading. An example includes the following question, found in Table 3:

Question 6. Was follow up complete and if not, were differences between groups in terms of their follow up adequately described and analyzed?

This question in particular requires to be separated in two paragraphs, because its reading is confusing and allows for bias in the exploration of data.

  1. The authors added the footnote of Figure 1 after Table 7, but such footnote does not correspond to the actual Figure 1. Could it be possible that the footnote found after Table 7 represents a possible Figure 2? If so, such figure was never found in the text nor described elsewhere. Is it missing? Should it be added and described?

  1. The way the Introduction section is written is confusing. Although showing first epidemiological data and then generalities of the clinical classification of the disease is an appropriate approach, the authors decided to briefly talk about diagnosis and treatment in the first paragraph together with epidemiology. Next, they explained symptoms of advanced pancreatic cancer, without first explaining what advancer pancreatic cancer is (it corresponds to stages III-IV, when cancer is unresectable). They did not comment anything about the symptoms and signs of the remaining types of pancreatic cancer. Next, they focused on treatment without providing information of the reasoning behind supporting gemcitabine without the whole FOLFIRINOX scheme. The authors have indicated that such approach is related with higher toxicity, but did not showed the secondary effects nor the references. In fact, they are not clearly indicating why gemcitabine and paclitaxel are superior to any other schemes or approaches in the treatment of advanced pancreatic cancer. Besides, is there any particular reason why they are not comparing normal paclitaxel plus gemcitabine?

  1. In Table 7, please add the description of the type of pancreatic cancer that is under study for each one of the molecules that are shown, and the effects that are seen when they are added in the treatment scheme.

  1. b) Minor comments

  1. The authors must ensure consistency in their manuscript. As an example, sometimes they talked about hypopotassemia (line 218), and sometimes they describe it as hypokalemia (Table 5). Additionally, sometimes they wrote “Nab-paclitaxel” (i.e., line 84), but in the title it is shown as “nab-paclitaxel”. It is requested to the authors to pay attention to these small mistakes and to respect their own acronyms.

  1. Please show the reference to support the next paragraph found in lines 53-54:

“…between the 50-55% of the patients suffer from metastatic disease while the 20% suffer from resectable disease (stages I and II)”.

Also, are the authors referring to patients at the moment of diagnosis? And is this data coming from a particular country or study?

  1. Please define Nab-paclitaxel first (line 84) and not in line 86.

  1. The question found in lines 100-101 implies that the authors focused in all the types of pancreatic cancer, but in lines 117-118 they stipulated that the eligibility criteria focused on advanced or metastatic pancreatic cancer. Therefore, it is asked to change the question in lines 100-101 to suit better their eligibility criteria.

  1. In Figure 1, please add the abbreviations in the footnote. Additionally, it would be useful if the authors could add a small subsection covering the acronyms and abbreviations used in the text.

  1. Please move the paragraph “Records have been included if they were approved by an ethics committee” found in line 160 into the Methods section.

  1. Please remove the repeated paragraph found in Table 5 about peripheral edema.

  1. Could you please add in the footnote of Table 6 the explanation of the acronyms, such as UTI?

Author Response

Dear Reviewer, thank you for reading our manuscript and for sending us your comments and suggestions. Our responses are given in a point-by-point manner below. We have highlighted the changes to our manuscript within the document by using colored text in MS Word.

1 a) Major comments

  1. The authors must request a professional proofreading service for English language, due to the multiple typos and errors found thorough the text. Besides, the way some paragraphs are written is very suggestive of a literal translation from the Spanish language, without taking into account the grammatical rules of English. Thus, the authors are required to submit a proof of such service together to their next manuscript version. Some typos include, but are not limited to:

  1. a) However (line 71). âœ”
  2. b) Neadjuvant (line 72). ✔
  3. c) The locally advanced or metastatic (line 74).
  4. d) , it (line 88). ✔
  5. e) Tolerate (line 166). ✔
  6. f) 74 patients participated (lines 166-167).
  7. g) Can, without the word ‘it’ before (line 205).
  8. h) Comparisons similar (Table 3).
  9. i) Edema peripheral (Table 5). ✔
  10. j) Neuropathy peripheral (Table 5). ✔
  11. k) Weight decreases, instead of loss (Table 5). ✔
  12. l) Alkaline phosphatase increase (Table 6). 
  13. m) An ‘of’ missed between ‘grades’ and ‘adverse’ (title of Table 6). ✔
  14. n) All the ‘it’ that are missing at the beginning of the background paragraphs in Table 7. ✔

Response: Thank you for taking the time to carefully read the manuscript. We corrected every point you listed, you'll find a checkmark next to each annotation, then the revised manuscript was sent to a professional proofreading service for English language. You will find the corrected version with the changes highlighted. In addition, we know that if the manuscript is accepted, it will go through the language edition again, this time by the editorial office, so the grammar and phrasing of the final version of the article will be improved.

Besides, some of the paragraphs are excessively long and required to be split in shorter paragraphs to avoid confusions in their reading. An example includes the following question, found in Table 3:

Question 6. Was follow up complete and if not, were differences between groups in terms of their follow up adequately described and analyzed?

This question in particular requires to be separated in two paragraphs, because its reading is confusing and allows for bias in the exploration of data.

Response: Thanks for your observation.

Question 6 was separated in two paragraphs to facilitate reading.

We have shortened the paragraphs in the Introduction, section 3.4, and Discussion to avoid confusions in their reading.

The order of the references has been updated.

  1. The authors added the footnote of Figure 1 after Table 7, but such footnote does not correspond to the actual Figure 1. Could it be possible that the footnote found after Table 7 represents a possible Figure 2? If so, such figure was never found in the text nor described elsewhere. Is it missing? Should it be added and described?

Response: We appreciate your observation. Indeed, in the original manuscript there was a Figure 2, but we decided to use it as a Graphical Abstract. In the revised manuscript we removed that footnote.

  1. The way the Introduction section is written is confusing. Although showing first epidemiological data and then generalities of the clinical classification of the disease is an appropriate approach, the authors decided to briefly talk about diagnosis and treatment in the first paragraph together with epidemiology.

Response: Thanks for your suggestion. We have restated the first paragraph showing epidemiology data and then generalities of the clinical classification of the disease. Thus, we have included the references:

  • Tempero, M.A.; et al. Pancreatic Adenocarcinoma, Version 2.2021, NCCN Clinical Practice Guidelines in Oncology. Journal of the National Comprehensive Cancer Network 2021, 19, 439–457, doi:10.6004/JNCCN.2021.0017.
  • Roalsø, M.; et al TNM-Staging for Pancreatic Adenocarcinoma – Real Progress or Much Ado about Nothing? European Journal of Surgical Oncology 2020, 46, 1554–1557, doi:10.1016/J.EJSO.2020.02.014.
  • Liu, C.; et al. Application of the Eighth Edition of the American Joint Committee on Cancer Staging for Pancreatic Adenocarcinoma. Pancreas 2018, 47, 742–747, doi:10.1097/MPA.0000000000001073.
  • Allen, P.J.; et al. Multi-Institutional Validation Study of the American Joint Commission on Cancer (8th Edition) Changes for T and N Staging in Patients with Pancreatic Adenocarcinoma. Annals of surgery 2017, 265, 185, doi:10.1097/SLA.0000000000001763.
  • Mostafa, M.E.; et al. Pathologic Classification of “Pancreatic Cancers”: Current Concepts and Challenges. Chinese clinical oncology 2017, 6, doi:10.21037/CCO.2017.12.01.
  • Puckett, Y.; Garfield, K. Pancreatic Cancer. StatPearls 2022

Next, they explained symptoms of advanced pancreatic cancer, without first explaining what advancer pancreatic cancer is (it corresponds to stages III-IV, when cancer is unresectable). They did not comment anything about the symptoms and signs of the remaining types of pancreatic cancer.

We have eliminated the second paragraph of the introduction because we reconsider that it is not focused on the main objective of the study. 

We have also removed the following text: Some of the symptoms of advanced pancreatic cancer are jaundice, abdominal pain, loss of appetite, pain in the back, epigastric pain, nausea, pancreatitis, involuntary weight loss, among others [29–31], for the same reason, and to avoid the Introduction being "too broad" as another reviewer stated.

Next, they focused on treatment without providing information of the reasoning behind supporting gemcitabine without the whole FOLFIRINOX scheme. The authors have indicated that such approach is related with higher toxicity, but did not showed the secondary effects nor the references. 

Response: We have included the following references

  • Lambert, A.; et al. Current Status on the Place of FOLFIRINOX in Metastatic Pancreatic Cancer and Future Directions. Therapeutic Advances in Gastroenterology 2017, 10, 631–645, doi:10.1177/1756283X17713879  (Although FOLFIRINOX was also associated with increased toxicity, mainly febrile neutropenia and diarrhea, there has...)
  • Moris, D.; et al. Initial Experience with Neoadjuvant FOLFIRINOX as First Line Therapy for Locally Advanced Pancreatic Cancer. JBUON 2020, 25, 2525–2527. (a. Roughly every patient experienced toxicity according to ECOG criteria..., and b. However, due to high toxicity, treatment is only feasible in selected patients and requires close monitoring...)

In fact, they are not clearly indicating why gemcitabine and paclitaxel are superior to any other schemes or approaches in the treatment of advanced pancreatic cancer.

Response: We have added the following text with its respective reference Furthermore, when the efficacy and safety of NP/G and FOLFIRINOX were compared, the response rate was shown to be 6.3% in the FOLFIRINOX group and 40.9% in the GnP and Drug toxicity in the NP/G group was less than in the FOLFIRINOX group [39].

Muranaka, T.; et al. Comparison of Efficacy and Toxicity of FOLFIRINOX and Gemcitabine with Nab-Paclitaxel in Unresectable Pancreatic Cancer. Journal of Gastrointestinal Oncology 2017, 8, 566–571, doi:10.21037/JGO.2017.02.02.

Besides, is there any particular reason why they are not comparing normal paclitaxel plus gemcitabine?

Response: The main reason is that in the regimens found in the literature, it is more common to find the nab-paclitaxel form. The reviewer will be able to verify it with this search in Pubmed:

https://pubmed.ncbi.nlm.nih.gov/?term=%28paclitaxel%29+AND+%28gemcitabine%29&sort=pubdate&size=100

  1. In Table 7, please add the description of the type of pancreatic cancer that is under study for each one of the molecules that are shown, and the effects that are seen when they are added in the treatment scheme.

Response: In all the cases presented in the Table 7, the pancreatic cancer under study is the advanced pancreatic ductal adenocarcinoma as mentioned in "2.2 Eligibility criteria" We included records of clinical trials that involve patients over 18 years old with advanced or metastatic pancreatic cancer. To avoid confusion, we made the research question more specific by mentioning the type of pancreatic cancer, and we changed the heading of Table 7. Likewise, in Table 7 we add a summary of the effects of the triple regimen and the identifier of each clinical trial.

  1. b) Minor comments

  1. The authors must ensure consistency in their manuscript. As an example, sometimes they talked about hypopotassemia (line 218), and sometimes they describe it as hypokalemia (Table 5). Additionally, sometimes they wrote “Nab-paclitaxel” (i.e., line 84), but in the title it is shown as “nab-paclitaxel”. It is requested to the authors to pay attention to these small mistakes and to respect their own acronyms.

Response: Thanks for your observation. We retain the term "hypokalemia" instead of "hypopotassemia". Also, 

We keep the term "nab-paclitaxel" with a lowercase n, except when it begins a sentence, as in: Nab-paclitaxel is a formulation of paclitaxel with albumin that is...  

  1. Please show the reference to support the next paragraph found in lines 53-54:

“…between the 50-55% of the patients suffer from metastatic disease while the 20% suffer from resectable disease (stages I and II)”.

Response: That paragraph was removed, as mentioned above. Anyway, the reference is:

  • Soweid, A.M. The Borderline Resectable and Locally Advanced Pancreatic Ductal Adenocarcinoma: Definition. Endoscopic Ultrasound 2017, 6, 76, doi:10.4103/EUS.EUS_66_17 (At initial presentation, the majority of patients (50%–55%) have metastatic disease while only 20% have resectable disease...)

This was raised in the first paragraph of the Introduction: Consequently, patients with previously untreated advanced pancreatic ductal adenocarcinoma, which represent 50-55% of cases [7], have a very short life expectancy

Also, are the authors referring to patients at the moment of diagnosis? And is this data coming from a particular country or study?

Response: Information on the nationality or other demographic information of the patients is not reported in the protocols of the clinical studies. The reviewer will be able to verify this in the protocols as in: https://clinicaltrials.gov/ProvidedDocs/68/NCT02436668/Prot_001.pdf  

  1. Please define Nab-paclitaxel first (line 84) and not in line 86.

Response: The definition of nab-paclitaxel was placed where you indicated, note that the line numbering has changed

  1. The question found in lines 100-101 implies that the authors focused in all the types of pancreatic cancer, but in lines 117-118 they stipulated that the eligibility criteria focused on advanced or metastatic pancreatic cancer. Therefore, it is asked to change the question in lines 100-101 to suit better their eligibility criteria.

Response: Thank you very much for making this important observation. Consequently, we have modified the research question to be consistent with the eligibility criteria.

  1. In Figure 1, please add the abbreviations in the footnote. Additionally, it would be useful if the authors could add a small subsection covering the acronyms and abbreviations used in the text.

Response: Abbreviations were added in the footnote of Figure 1. The "Abbreviations" subsection was added in the revised manuscript.

  1. Please move the paragraph “Records have been included if they were approved by an ethics committee” found in line 160 into the Methods section.

Response: "Records have been included if they were approved by an ethics committee" was moved to subsection 2.2 Eligibility criteria

  1. Please remove the repeated paragraph found in Table 5 about peripheral edema.

Response: The repeated paragraph was removed

  1. Could you please add in the footnote of Table 6 the explanation of the acronyms, such as UTI?

Response: The acronyms were added in the footnote of Table 6. In addition, they were included in the Abbreviations and acronyms subsection.

Once again, we thank you for taking the time to carefully read our manuscript and for making all these observations. The revised manuscript was sent to a professional proofreading service for English language. You will find the corrected version with the changes highlighted. We hope that our responses are satisfactory and that you find the revised version of the manuscript suitable for publication.

Sincerely, 

Reviewer 4 Report

The manuscript covers a very important subject regarding the clinical management of PDAC patients. In latest years, the treatment of non-resectable PDAC has not seen significant progress therefore it is of the utmost importance to understand if the use of additional drugs in a combinatory fashion to what is currently used brings any advantage to cancer patients. Even though the subject is well covered to a certain extent, in many instances sentences are not properly written which makes it difficult for the information to be drawn. Please find my questions and comments below:

Question 1: The authors of the manuscript aimed to update the literature regarding the clinical advantages of the use of the NP/G regimen in combination with one of 7 additional drugs known to have an anti-tumorigenic effect. Thus, a clear explanation of the biological effects of these drugs together with examples of their application is missing and should therefore be added.  

Question 2: The number of studies used in the end for this manuscript is very reduced (mostly if only 4 of them were done with the proper control group). In figure 1 it is stated that 799 records were excluded due to the lack of access which seems like an excessive number. What were the main reasons for this?  

All manuscript: Abbreviations (such as PDAC) should be used consistently throughout the manuscript.

Line 33-34: Please rephrase by indicating the ranking of PDAC in terms of incidence and mortality for both genders worldwide.

Line 36-40: Please rephrase. The structure of the sentence is confusing and cannot be understood properly.

Line 41-45: Please rephrase. The structure of the sentence is confusing and cannot be understood properly.

Line 60-64: Sentence is too long. Please divide it in 2 independent sentences.

Line 74-78: Please rephrase. The structure of the sentence is confusing and cannot be understood properly.

Line 95: Please add reference at the end.

Line 101: Replace “has” with “have”

Line 128: Replace “less than” with “before”: Also correct “2005” to “2015” as stated in the criteria from figure 1.

Line 142: Please replace “in” with “of”

Line 154: correct sentence

Line 220: Full stop at the end of sentence. New sentence starting in the beginning of line 221

Line 231: Correct “reception” with “receptor”

Line 252: The numbers at the beginning of a sentence should be written with words and not numbers.

Line 258-260: Meaning of the sentence is not clear. Do the authors wanted to say that the molecular effects of the mentioned drugs are well characterized in cancer cells?

Line 262-265: Sentence is confusing, please rephrase.

Line 270: The authors refer pathways in the figure legend but there are no pathways shown in the figure per se. Please rephrase. Also this corresponds to figure 2 of the manuscript and not figure 1.

Line 301: Please correct IMPACT name

Line 314: Please elaborate what RECIST is.

Author Response

Dear Reviewer, thank you for reading our manuscript and for sending us your comments and suggestions. Our responses are given in a point-by-point manner below. We have highlighted the changes to our manuscript within the document by using colored text in MS Word.

Question 1: The authors of the manuscript aimed to update the literature regarding the clinical advantages of the use of the NP/G regimen in combination with one of 7 additional drugs known to have an anti-tumorigenic effect. Thus, a clear explanation of the biological effects of these drugs together with examples of their application is missing and should therefore be added. 

Response: The biological effects of gemcitabine and nab-paclitaxel were discussed in the Introduction, while the biological effects and some examples of their application of the 7 therapeutic agents are summarized in Table 7. 

References added:

  • MacDonald, A.; et al. Necuparanib, A Multitargeting Heparan Sulfate Mimetic, Targets Tumor and Stromal Compartments in Pancreatic Cancer. Molecular Cancer Therapeutics 2019, 18, 245–256, doi:10.1158/1535-7163.MCT-18-0417.
  • Ibrutinib: Uses, Interactions, Mechanism of Action | DrugBank Online Available online: https://go.drugbank.com/drugs/DB09053 (accessed on 16 January 2022).

  • Hughes, D.L. Patent Review of Manufacturing Routes to Recently Approved Oncology Drugs: Ibrutinib, Cobimetinib, and Alectinib. Organic Process Research and Development 2016, 20, 1855–1869, doi:10.1021/ACS.OPRD.6B00304.
  • MacDonald, A.; et al. Necuparanib, A Multitargeting Heparan Sulfate Mimetic, Targets Tumor and Stromal Compartments in Pancreatic Cancer. Molecular Cancer Therapeutics 2019, 18, 245–256, doi:10.1158/1535-7163.MCT-18-0417.
  • Shah, J.; et al. Tarextumab (Anti-NOTCH2/3) Reverses NOTCH2 and NOTCH3-Dependent Tumorigenicity and Metastases in Small Cell Lung Cancer. Cancer Research 2015, 75, 2323–2323, doi:10.1158/1538-7445.AM2015-2323.
  • Crooke, S.T.; et al. Antisense Technology: An Overview and Prospectus. Nature Reviews Drug Discovery 2021 20:6 2021, 20, 427–453, doi:10.1038/s41573-021-00162-z. 
  • Ghosh, S. Cisplatin: The First Metal Based Anticancer Drug. Bioorganic Chemistry 2019, 88, 102925, doi:10.1016/J.BIOORG.2019.102925.
  • Erdogan, B.; et al. Enzalutamide in Prostate Cancer, A Review on Enzalutamide and Cancer. Eurasian Journal of Medicine and Oncology 2018, 2, 121–129, doi:10.14744/EJMO.2018.72098.
  • Azhar, M.; et al. Momelotinib Is a Highly Potent Inhibitor of FLT3-Mutant AML. Blood Advances 2021, doi:10.1182/BLOODADVANCES.2021004611.

Question 2: The number of studies used in the end for this manuscript is very reduced (mostly if only 4 of them were done with the proper control group). In figure 1 it is stated that 799 records were excluded due to the lack of access which seems like an excessive number. What were the main reasons for this?  

Response: It is common that in systematic reviews the initial search begins with a large number of records and only a few are included in the end, as in our work

For example: 

  • https://systematicreviewsjournal.biomedcentral.com/track/pdf/10.1186/s13643-021-01820-4.pdf
    • In this case, 1715 records were identified, 1517 were excluded, and only 5 studies were included in the review
  • https://systematicreviewsjournal.biomedcentral.com/track/pdf/10.1186/s13643-020-01425-3.pdf

    • In this case, 4316 records were identified, 2410 were excluded, and only 9 studies were included in the review 

In our research, 799 were excluded records for two main reasons, the first is when reading the title and abstract it did not fit with the approach of the review, and the second is that we could not access the content using the tools with which we legally have such as EndNote Click (Clarivate), Mendely Web Importer (ELSEVIER), or by subscription of our institutions. Thanks for your appreciation.

All manuscript: Abbreviations (such as PDAC) should be used consistently throughout the manuscript.

Response: We made sure that the abbreviations and acronyms were consistent, we even added a small subsection that compiles the abbreviations and acronyms mentioned in the manuscript

Line 33-34: Please rephrase by indicating the ranking of PDAC in terms of incidence and mortality for both genders worldwide.

Response: The sentence was rephrase. The global incidence and mortality for both sexes were indicated.

Line 36-40: Please rephrase. The structure of the sentence is confusing and cannot be understood properly.

Response: The sentence was rephrase.

Line 41-45: Please rephrase. The structure of the sentence is confusing and cannot be understood properly.

Response: The sentence was rephrase.

Line 60-64: Sentence is too long. Please divide it in 2 independent sentences.

Response: The sentence was divided to improve reading

Line 74-78: Please rephrase. The structure of the sentence is confusing and cannot be understood properly.

Response: The sentence was rephrase.

Line 95: Please add reference at the end.

Response: We add the reference at the end of the paragraph

  • Giordano, G.; et al Nano Albumin Bound-Paclitaxel in Pancreatic Cancer: Current Evidences and Future Directions. World Journal of Gastroenterology 2017, 23, 5875–5886, doi:10.3748/wjg.v23.i32.587 (This agent is prepared by homogenization of human serum albumin at 3%-4% concentration with paclitaxel...)

Line 101: Replace “has” with “have”

Response: Replaced 

Line 128: Replace “less than” with “before”: Also correct “2005” to “2015” as stated in the criteria from figure 1.

Response: Thank you for your notes. We have already corrected this that you indicated

Line 142: Please replace “in” with “of”

Response: Thank you for your observation. We already attended this indication.

Line 154: correct sentence

Response: The sentence was corrected

Line 220: Full stop at the end of sentence. New sentence starting in the beginning of line 221

Response: We have already addressed this observation. Thank you again

Line 231: Correct “reception” with “receptor”

Response: Thank you for your observation. We have already corrected this.

Line 252: The numbers at the beginning of a sentence should be written with words and not numbers.

Response: Thank you for your observation. We have already written with letters instead of numbers at the beginning of the sentence.

Line 258-260: Meaning of the sentence is not clear. Do the authors wanted to say that the molecular effects of the mentioned drugs are well characterized in cancer cells?

Line 262-265: Sentence is confusing, please rephrase.

Line 270: The authors refer pathways in the figure legend but there are no pathways shown in the figure per se. Please rephrase. Also this corresponds to figure 2 of the manuscript and not figure 1.

Response: We appreciate your observation. In the original manuscript there was a Figure 2, but we decided to use it as a Graphical Abstract. In the revised manuscript we removed that footnote.

Line 301: Please correct IMPACT name

Response: The correct name is MPACT, without the letter I. The reviewer can confirm it by clicking the following link

https://doi.org/10.1093/annonc/mdt201.1

Line 314: Please elaborate what RECIST is.

Response: We include the following text to address your observation: Therefore, confirmation of this clinical finding should be performed by Response Evaluation Criteria in Solid Tumors (RECIST), which provides a simple and pragmatic methodology to evaluate the activity and efficacy of new cancer therapeutics in solid tumors, using validated and consistent criteria to assess changes in tumor burden 

Added References:

  • Therasse, P.; et al. New Guidelines to Evaluate the Response to Treatment in Solid Tumors. JNCI: Journal of the National Cancer Institute 2000, 92, 205–216, doi:10.1093/JNCI/92.3.205.
  • Litière, S.; et al. RECIST 1.1 for Response Evaluation Apply Not Only to Chemotherapy-Treated Patients but Also to Targeted Cancer Agents: A Pooled Database Analysis. Journal of Clinical Oncology 2019, 37, 1102–1110, doi:10.1200/JCO.18.01100.

Once again, we thank you for taking the time to carefully read our manuscript and for making all these observations. The revised manuscript was sent to a professional proofreading service for English language. You will find the corrected version with the changes highlighted. In addition, we know that if the manuscript is accepted, it will go through the language edition again, this time by the editorial office, so the grammar and phrasing of the final version of the article will be improved. We hope that our responses are satisfactory and that you find the revised version of the manuscript suitable for publication.

Sincerely, 

Round 2

Reviewer 1 Report

In the light of the revised manuscript, I have no further comments

Reviewer 3 Report

The authors have replied diligently to each one of the questions and comments from the previous report, and now the manuscript is ready for acceptance.